# Urban Climate Vulnerability in Cambodia: A Case Study in Koh Kong Province

**Kimleng Sa** [iD]

Faculty of Development Studies, Royal University of Phnom Penh, Phnom Penh 12105, Cambodia; sakimleng@yahoo.com; Tel.: +855-92-276-076

**Abstract:** This study investigates an urban climate vulnerability in Cambodia by constructing an index to compare three different communes, Smach Meanchey, Daun Tong, and Steong Veng, located in the Khemarak Phoumin district, Koh Kong province. It is found that Daun Tong commune is the most vulnerable location among the three communes, followed by Steong Veng. Besides, vulnerability as Expected Poverty (VEP) is used to measure the vulnerability to poverty, that is, the probability of a household income to fall below the poverty line, as it captures the impact of shocks can be conducted in the cross-sectional study. It applies two poverty thresholds: the national poverty line after taking into account the inflation rate and the international poverty line defined by the World Bank, to look into its sensitivity. By using the national poverty line, the study reveals that more than one-fourth of households are vulnerable to poverty, while the international poverty threshold shows that approximately one-third of households are in peril. With low levels of income inequality, households are not highly sensitive to poverty; however, both poverty thresholds point out that the current urban poor households are more vulnerable than non-poor families.

**Keywords:** vulnerability; urban; index; vulnerability as expected poverty; shocks

**JEL Classification:** O18; Q54

## 1. Introduction

Climate change is becoming a crucial problem for city dwellers, especially for urban poor in low-income countries. Informal settlements become a common practice for urban poor because they cannot afford good-quality housing (McGranahan et al. 2007). They are strongly threatened by environmental changes since they are more sensitive to the dynamics of natural resources and lack the means to improve their adaptive capacity to natural hazards (Satterthwaite et al. 2007). Human settlements located along coastal and river floodplains are the most vulnerable areas of climate change, as their economic activities highly depend on climate-sensitive resources, especially during the dry season when water sanitation cannot supply the whole population.

Urban adaptation to climate change is often linked to the role of governance; however, official municipal policies frequently increase the vulnerabilities—including social, economic, and environmental vulnerabilities—of poor residents rather than reduce them (Satterthwaite et al. 2007). Recent urbanization has induced climate-related problems that disrupt the economic activities and livelihoods of Cambodians in urban areas. A rapid population growth in the city has increased the demand for water and electricity; on the other hand, a low governance capacity to supply these needs pushes the prices up and increases the cost of living, which mainly affects poor people in the city. Households living in the slum area often face health problems. UNICEF (2010) reported a persistent inequality in access to healthcare services and health threats due to communicable diseases in a poor community in Phnom Penh.

In Cambodia, the loss and damage caused by climate change are increasing and interrupting sustainable development. According to Ancha Srinivasan,[1] economic loss due to climate change rose to approximately 10% of the GDP in 2015. Hazards, mainly floods, resulted in a GDP loss about 4.3% in 2011, as reported in the International Disaster Database (MoE 2013). From 1984 until 2011, Cambodia encountered flooding 20 times (Chhinh 2014, p. 168).

The Royal Government of Cambodia identifies the coastal zone as the most vulnerable location of climate change. Recent urbanization due to economic growth has posed many challenges not only for people living in Phnom Penh, but also for residents along the coastal zones where economic activity has been expanding substantially. These problems include waste and sanitation, water shortage, water quality (Irvine et al. 2006), and energy security (Heng 2012). Moreover, urban settlement along the coastal zone has been exposed to climate hazards such as storm surges, cyclones, seawater intrusion, and other water stresses (Hay and Mimura 2006; McGranahan et al. 2007).

The coastal zone of Cambodia is comprised of four provinces: Koh Kong, Sihanoukville, Kampot, and Kep. Koh Kong is a southwestern province in Cambodia that plays an important role as an economic corridor. First, it serves as an economic gateway that links Thailand and Cambodia, as well as access to the main port of Sihanoukville. Second, this Special Economic Zone (SEZ) absorbs a large amount of labor to work in garment factories and manufacturing plants in marine production. Third, it possesses many eco-tourism sites that are the important agents to improve the livelihood of the local residents. Since local people have depended strongly on the natural environment, a slight change will generate a significant impact on their income and livelihood.

A large part of Koh Kong has been threatened by climate hazards. As it is situated in the coastal area, it has been endangered by climate shocks such as storm surges, droughts, floods, and seawater intrusion, especially affecting low-income citizens who hold little capacity to cope with it. According to UNEP (2013), the rainfall along the coastal area was predicted to rise 2% to 6% by 2050. Heavy rain in the rainy season has induced the risk of storm and flash flooding in the low-lying area where agricultural crops are mostly concentrated. Climate Investment Funds (CIF, p. 48) projected that in the worst-case scenario, the annual mean temperature was expected to increase by 1 °C by 2025 in Koh Kong. In addition, a study of MoE (2002) conducted in Peam Krasoab, a commune in Koh Kong, found that a 1-m rise of seawater would cause 44 square kilometers (about 0.4% of Koh Kong) to lie under water permanently.

Since climate hazards strongly affect middle- and low-income citizens, the study aims to examine the vulnerability of climate shocks on urban residents in Koh Kong province. The two main objectives of the study are listed as follows:

1. To identify the most vulnerable region of climate hazards through the construction of an index to compare three different communes in Koh Kong.
2. To measure the vulnerability to poverty in urban locations through the comparison of poor and non-poor households.

## 2. Methodology

### 2.1. Study Area

According to the national census in 2008, Koh Kong is divided into eight districts with 33 communes. This study was conducted in the Smach Meanchey district, where the provincial town of Koh Kong is located, which later was changed to Khemarak Phoumin district. It is comprised of three communes: Smach Meanchey, Daun Tong, and Steong Veng. The population of Smach Meanchey was about 29,329 which was 21% of the total population in Koh Kong, according to national census in 2008. Base on the national census (1998) conducted by the National Institute of Statistics (NIS), an urban area

---

[1]　Ancha Srinivasan is ADB's climate change specialist of Southeast Asia Department.

refers to any district in which a provincial town is located. With a clarification of the term "urban" due to population growth, the new definition of "urban" was adopted in 2004 and was used in the national census in 2008 based on three categories: population density exceeding 200 per Km$^2$, percentage of males employed in the agricultural sector lower than 50%, and the total population of a commune exceeding 2000 (NIS 2008).

Figure 1 shows the location of Smach Meanchey district. It is located in the western part of Koh Kong. Although Daun Tong is relative small compared to the other two communes, high population density concentrates in this commune.

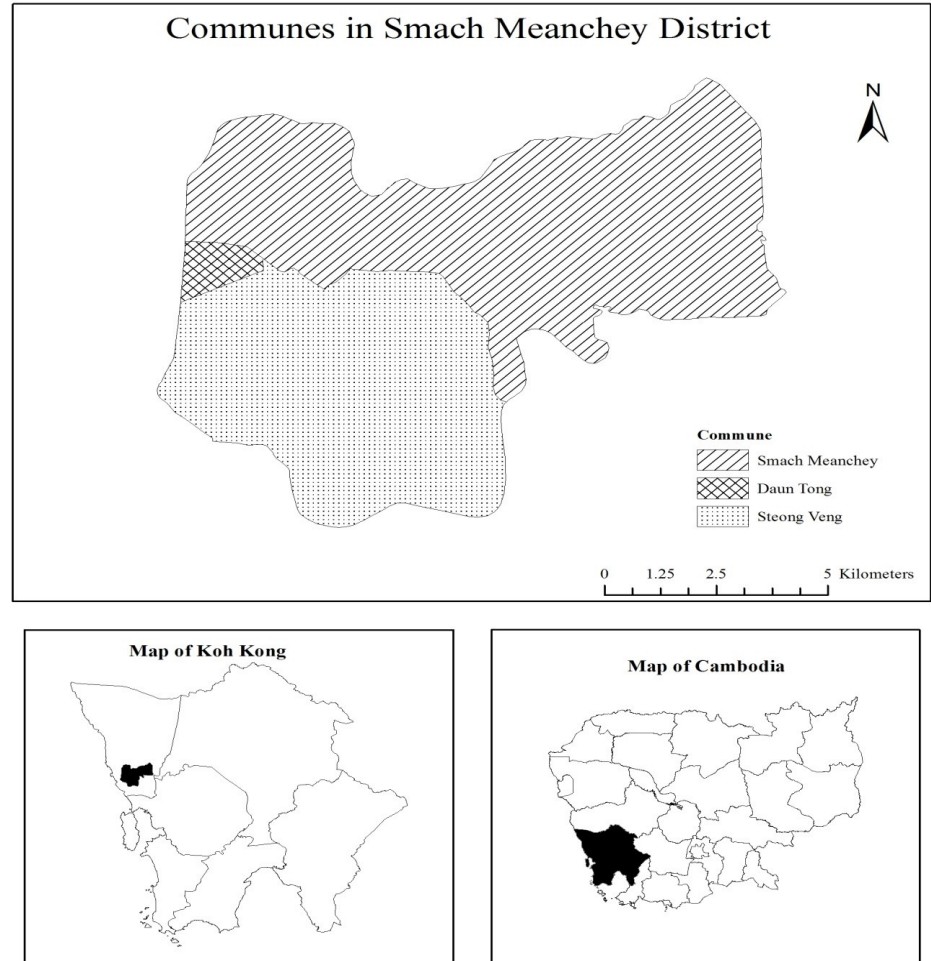

**Figure 1.** Map of the study area.

## 2.2. Sampling Methodology

This study applied a quantitative method by using a semi-structured questionnaire for the household interview during October 2016. It used the nonprobability sampling method and followed the steps below in choosing the respondent.[2]

First, the location was chosen based on its geographical condition. In terms of geography, it selected the villages in each commune according to their history of natural hazards. It only chose the

---

2     The study used purposing sampling based on researcher knowledge to identify the participants, so the sample may not truly represent the population due to the lack of randomness. Thus, the results of this study may not be able to represent the urban vulnerability in other regions.

villages where hazards tended to be extreme and frequently occurred over a long period, compared to the other villages in the commune. Since the study simply selected the places where hazards used to exist, this may have resulted in a biased behavior in the data collection; however, it also enhanced the result to be more accurate for comparison. Next, the number of samples in each commune was selected based on the proportion of households in each commune compared to the total population. In total, 120 households were interviewed in Smach Meanchey, Daun Tong, and Steong Veng communes with the sample sizes of 50, 33, and 37, respectively. Lastly, it selected households via the distance from one to another, about 200 m. It targeted the head of the household to be a respondent.

For ethical and confidential considerations, consent was requested from every participant. If a participant was unwilling to join the interview, the interviewer would ask for any available member in the family who could represent the household; otherwise, the interviewer would try to access the neighboring household as a substitution. Moreover, the interview did not use any recording device so as to make household feel comfortable. All the information provided only was used for study and was destroyed after six months.

*2.3. Vulnerability Index*

The study followed the definition of the Intergovernmental Panel on Climate Change (IPCC), which identified vulnerability as "the degree to which a system is susceptible to, or unable to cope with, adverse effects of climate change, including climate variability and extremes" (McCarthy et al. 2001, p. 6). It viewed vulnerability as the function of exposure, sensitivity, and adaptive capacity that is widely recognized in the study of climate vulnerability assessment.

Measuring vulnerability is a challenge, and mostly proxy variables are used to construct an index in order to compare the vulnerability degree across different regions. Srinivasan et al. (2013) proposed the need to create a capable governance to reduce urban vulnerability to water shortage. Depietri et al. (2013) assessed urban vulnerability to heat waves in urban Germany via GIS with the use of a series of social and ecological indicators such as resource, infrastructure, settlement, health, death rate, and so on. Chen et al. (2013) studied vulnerability to natural hazards in China by using the PCA approach and found that employment and poverty, education, housing quality, size of family, minority, and housing size were the significant determinants of vulnerability. Andersen and Cardona (2013) introduced the Livelihood Diversification Index (LDI) to study vulnerability and resilience capacity in Bolivia. Hahn et al. (2009) developed the Livelihood Vulnerability Index (LVI) to assess vulnerability in Mozambique. Ncube et al. (2016) constructed the Household Vulnerability Index (HVI) to study the vulnerability of climate change in South Africa. Basically, it estimated the household vulnerability according to the natural assets, physical assets, financial assets, human capital assets, and social assets derived from Sustainable Livelihood Index (Solesbury 2003). The use of indicators to construct an index in the study came from the review of the past studies that were recognized for their significant contribution in vulnerability studies.

A weighing indicator was one of the challenges in constructing the index. Some studies used equal weight, as in Hahn et al. (2009), while some preferred to consult with an expert, as in Chhinh and Cheb (2013). The most common method for the weighing indicator was Principal Component Analysis (PCA), as recommended in (Gbetibouo and Ringler 2009; Piya et al. 2012; Megersa 2015). This paper used the PCA approach to assign weight. A drawback of using PCA was the correlation sign produced by PCA; however, the problem was not serious, and it was recommended to ignore the issue. This was because changing the correlation sign in the component did not change the variance, so it did not affect the weighing of the variable. Although the sign was not the issue in the mathematical explanation, the study preferred to consider the second or third principal components as well when the eigenvalue was similar to the first principal component and produce better correct signs of the indicators, conforming to the past literature.

The first step in calculating the index is to normalize the value of variables by using the following formula:

$$Normalized\ Value = \frac{Observed\ Value - Minimum\ Value}{Maximum\ Value - Minimum\ Value}$$

Normalization was used to create uniform data sets, to make them comparable by scale measurement. In the study, the process transformed the values to range from 0 to 1, where the higher the value, the stronger the effect.

After the values were normalized, the weighing process was conducted. The weighing variable aimed to identify the relative importance of each variable among others in explaining a certain phenomenon. A review of the literature revealed three methods of weighing the indicators: expert judgment, equal weight, and econometric approaches such as principal components or factor analysis (Gbetibouo et al. 2010). The use of expert judgment was constrained due to the knowledge in different fields, and it was difficult to reach consensus among the experts. Some studies used equal weight; however, by doing so it underestimates some important variables and overestimates some unimportant variables. To avoid these problems, PCA was used in this study to produce the weight of the variables. The component score coefficient from the first PCA was used as a weight since it explained the highest variability; even so, the second and third component scores were also considered if their eigenvalues and the variabilities explained were comparable to the first component. By doing so, it was helpful to correct the wrong sign of the correlation produced by the first principal component.

Next, variables were added together to form the index based on their own categories such as exposure, sensitivity, and adaptive capacity by using the formula below:

$$I_{ij} = \sum_{i=1}^{n} b_i \left\{ \frac{x_{ij} - x_i^{min}}{x_i^{max} - x_i^{min}} \right\}$$

where $I_{ij}$ was an index value of the variable $i$ in $j$ category, $b_i$ was the weight received from the principal component—generally the first component (PCA1), $x_{ij}$ was the value of the variable $i$ in $j$ category, $x_i^{min}$ was the minimum value of variable $i$, and $x_i^{max}$ was the maximum value of variable $i$.

Lastly, the climate vulnerability index was calculated by the following formula:[3]

$$VI = \sqrt[3]{EI \times SI \times (1 - ACI)}$$

where *VI*: vulnerability index; *EI*: exposure index; *SI*: sensitivity index; and *ACI*: adaptive capacity index.

### 2.4. Vulnerability as Expected Poverty

Vulnerability is a concept that links to poverty, as it may put those who are not currently poor in poverty in the future. This idea has brought insight into the study of the wellbeing loss due to shocks, ex-post assessment. In this context, vulnerability can be explained as a situation in which shocks may cause the household income to fall below a certain threshold.

The study of vulnerability in the context of poverty explains the level of household capacity to sustain a certain shock and the dynamic of household livelihood conditions. Just because a household is currently poor does not mean that they are vulnerable and vice versa. The lack of means to smooth expenditure over time has instigated the notion of studying vulnerability as expected poverty (VEP).

In this study, both the urban national poverty line and international poverty line were used. The reason for using two difference poverty thresholds is to understand their sensitivities. As the

---

[3]　The study used geometric mean instead of arithmetic mean.

Cambodian national poverty line was calculated based on the prices in 2009, it will be adjusted according to inflation to determine the prices in 2015. It was used as a baseline, while the international poverty line was used to observe the rate of change from the baseline.

This study used the concept of Vulnerability as Expected Poverty (VEP) developed by Shubham Chaudhuri to study the vulnerability in the context of poverty. He identified vulnerability as the probability that the household income or consumption may fall below the poverty line due to shocks such as climate hazards or financial crisis, and so on, as expressed by this formula:

$$\ln C_h = X_h \beta + e_h \tag{1}$$

$\ln C_h$ represents daily income in log form;[4] $X_h$ is the bundle of household characteristics and climate exposure index; $\beta$ is a vector of parameters; and $e_h$ represents the disturbance that captures idiosyncratic factors (shocks).

Another assumption was that the variance $e_h$, which captures idiosyncratic factors and reflects the inter-temporal variance of income, is provided by Equation (2). Since $e_h$ could not be observed, the solution was to examine from the sample.

$$\sigma_{e,h}^2 = X_h \theta \tag{2}$$

Chaudhuri et al. (2002) expressed that vulnerability to poverty was not a linear function, since it depended not only on the mean of income level but also inter-temporal variance stemming from income; thus, it involved not only the prediction of future income but also the expected disturbance as well. Since it was assumed that the disturbance term possesses heteroscedasticity in nature, it will not produce efficient estimators. To resolve this issue, Amemiya (1977) carried out three-step Feasible Generalized Least Square (FGLS) to transform variables by assigning weight to produce efficient estimators: $\beta$ and $\theta$. To reduce the methodological problems, the data was checked before processing. The dependent variable was checked for normality and was transformed into a log form to avoid outliers. For explanatory variables, they were checked for multicollinearity by using a correlation matrix. If the correlation is higher than 70%, one variable is dropped out of the equation. As expected from the assumption of ramdom error, the residual $e_h$ is not normally distributed that created the non-constant variance, so weight was used to transform the data as a remedial measure.

Starting with Equation (1), an OLS regression was run to find the estimated residual $e_h$, then the square of estimated residual was used to regress on $X_h$ through the OLS procedure again.

$$\hat{e}_{OLS,h}^2 = X_h \theta + \mu_h \tag{3}$$

Then, the predicted value was used from the regression as weight, and $X_h \hat{\theta}$ was transformed from Equation (3) into:

$$\frac{\hat{e}_{OLS,h}^2}{X_h \hat{\theta}_{OLS}} = \left( \frac{X_h}{X_h \hat{\theta}_{OLS}} \right) \theta + \frac{\mu_h}{X_h \hat{\theta}_{OLS}} = X_h \hat{\theta}_{FGLS} + u_i \tag{4}$$

$X_h \hat{\theta}_{FGLS}$ was the consistent estimated variance in Equation (2), shown above. Through the FGLS procedure, this variance $\sigma_{e,h}^2$ was unbiased and can be written as a standard deviation as follows:

$$\hat{\sigma}_{e,h} = \sqrt{X_h \hat{\theta}_{FGLS}} \tag{5}$$

At this stage, the issue that some estimated values are not always positive may occur, so another procedure may need to be used, such as a logistic specification that would force the predicted value to

---

[4] As the study focused on the urban area, it was more appropriate to assume that households saved money, so income was used rather than consumption for its nature of heteroscedasticity.

always be positive.[5] However, in this study, we will ignore this issue and simply drop those data from the estimation. Subsequently, this standard deviation was employed to transform Equation (1) into:

$$\frac{lnC_h}{\sqrt{X_h\hat{\theta}_{FGLS}}} = \frac{X_h}{\sqrt{X_h\hat{\theta}_{FGLS}}}\beta + \frac{e_h}{\sqrt{X_h\hat{\theta}_{FGLS}}} \tag{6}$$

The OLS estimation from Equation (6) will produce a consistent and efficient $\beta$. To be sure, the residual test, the Shapiro-Wilk test, was used to check the normality. Finally, by using $\beta$ and $\theta$, the expected log of income and the variance of log income of each household were predicted as:

$$\hat{E}(lnC_h|X_h) = X_h\hat{\beta} \tag{7}$$

$$\hat{V}(lnC_h|X_h) = \hat{\sigma}^2_{e,h} = X_h\hat{\theta} \tag{8}$$

Assuming income as log-normal distributed, the above equation can estimate the probability that a household with characteristic $X_h$ will be poor (household vulnerability level). Let $\Phi$ refer to the cumulative density of the standard normal; the estimated probability equation is as follows:

$$\hat{V} = \hat{Pr}(lnC_h < lnz|X_h) = \Phi\left(\frac{lnz - X_h\hat{\beta}}{\sqrt{X_h\hat{\theta}}}\right) \tag{9}$$

where $lnz$ is the natural log of minimum income; at this level a household will be considered vulnerable (at or below the poverty line). $X_h\hat{\beta}$ was the expected mean of household income, while $X_h\hat{\theta}$ was the predicted variance.

## 3. Results

### 3.1. Household Characteristics

Table 1 summarizes the basic characteristics of the households. On average, the head of household was aged around 44 years old with five years of education. Normally, a household had five members with two people in the family earning income, and there was at least one person in a family who had finished secondary school on average. Average monthly household income and consumption were about 464 dollars/month and 388 dollars/month, respectively. Daily income and consumption per person were around 3.3 dollars/day and 2.75 dollars/day on average. There was a small difference between the income and consumption levels that reflected the lower rate of saving.

**Table 1.** Household characteristics.

| Household Characteristics | Total | | Daun Tong | | Steong Veng | | Smach Meanchey | |
|---|---|---|---|---|---|---|---|---|
| | Mean | Standard Deviation | Mean | Standard Deviation | Mean | Standard Deviation | Mean | Standard Deviation |
| Age of household head | 43.93 | 13.40 | 43.80 | 13.50 | 45.60 | 13.90 | 42.70 | 13.10 |
| Education of head of household | 4.91 | 4.00 | 4.30 | 3.60 | 4.60 | 4.10 | 5.50 | 4.20 |
| Household sizes | 5.04 | 2.14 | 5.70 | 2.10 | 5.00 | 2.40 | 4.70 | 1.90 |
| Household members who finished grade 9 | 1.07 | 1.19 | 1.50 | 1.30 | 0.70 | 1.20 | 1.00 | 1.10 |
| Household members who earn revenue | 2.13 | 1.06 | 2.40 | 1.10 | 2.10 | 1.10 | 1.90 | 1.00 |

---

5　See: Elbers, Lanjouw, and Lanjouw (Elbers et al. 2001).

**Table 1.** *Cont.*

| Household Characteristics | Total | | Daun Tong | | Steong Veng | | Smach Meanchey | |
|---|---|---|---|---|---|---|---|---|
| | **Mean** | **Standard Deviation** | **Mean** | **Standard Deviation** | **Mean** | **Standard Deviation** | **Mean** | **Standard Deviation** |
| Monthly household income (dollars) | 463.93 | 244.47 | 518.39 | 183.78 | 443.83 | 263.71 | 442.85 | 263.12 |
| Monthly household consumption (dollars) | 387.55 | 176.53 | 453.03 | 153.10 | 356.96 | 171.27 | 367.02 | 186.53 |
| Daily income per person (dollars) | 3.28 | 1.68 | 3.37 | 1.64 | 3.12 | 1.38 | 3.32 | 1.92 |
| Daily consumption per person (dollars) | 2.75 | 1.15 | 2.90 | 1.22 | 2.60 | 1.16 | 2.71 | 1.09 |

### 3.2. Gini Coefficient and Inequality

Monthly incomes of 120 households were collected to construct the Lorenz curve to measure income inequality in the Smach Meanchey district. Using the data from this survey, the Gini coefficient or Gini index was calculated. This index ranges from 0 to 1, where a higher value represents greater inequality. According to the data, the Gini index was around 0.28 (see Figure 2), so it can be considered as an indication of low inequality among households in the Smach Meanchey district. With low-income inequality, it would be useful to the study of the degree of the sensitivity of vulnerability to poverty as the assumption of income indifference across households. However, the lowest 10% controlled only about 3% of the total income, while the top 10% occupied almost one-fourth of the entire income.

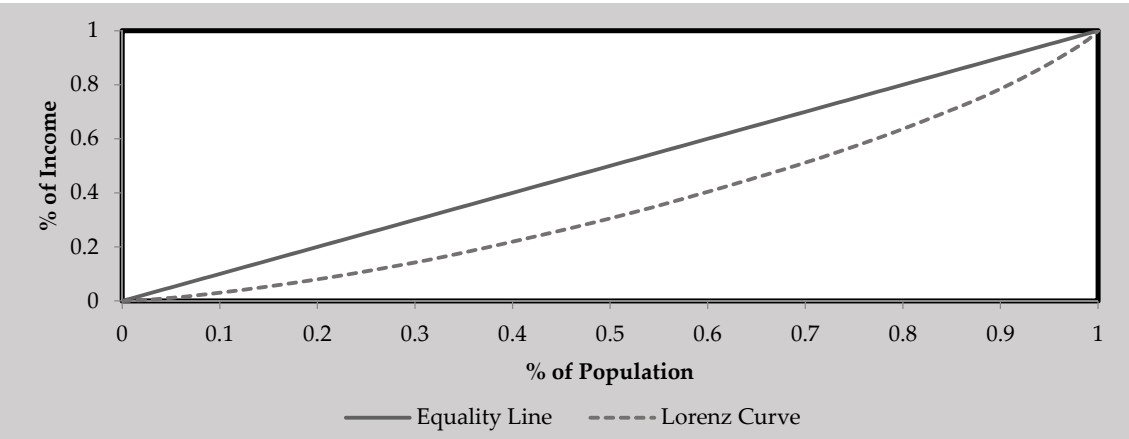

**Figure 2.** Income inequality in the Smach Meanchey district.

### 3.3. Climate Vulnerability

#### 3.3.1. Exposure Index

The rate of exposure was relatively stronger in Steong Veng and Daun Tong communes compared to Smach Meanchey, with indices of 0.456, 0.450, and 0.289, respectively (Table 2). PCA revealed the relative importance of storms and typhoons in explaining the exposure degree on household livelihood in the three communes compared to flooding and drought, as large component scores were contributed by these variables.

**Table 2.** Summary of exposure index.

| Descriptive | Component Score Coefficient | Daun Tong | Steong Veng | Smach Meanchey |
|---|---|---|---|---|
| | | Index | | |
| The number of floods affecting households that occurred during the last 5 years | 0.039 | 0.002 | 0.004 | 0.004 |
| Impact of flooding on livelihood | 0.107 | 0.029 | 0.040 | 0.041 |
| Period of insufficient clean water usage per year during the last 5 years | 0.103 | 0.027 | 0.034 | 0.038 |
| Impact of drought on livelihood | 0.043 | 0.032 | 0.029 | 0.027 |
| Frequency of typhoons affecting household livelihood per year during the last 5 years | 0.286 | 0.062 | 0.063 | 0.035 |
| Impact of typhoons on livelihood | 0.289 | 0.168 | 0.133 | 0.099 |
| Frequency of storms affecting household livelihood per year during the last 5 years | 0.312 | 0.036 | 0.053 | 0.016 |
| Impact of storms on livelihood | 0.329 | 0.094 | 0.099 | 0.028 |
| Total | | 0.450 | 0.456 | 0.289 |
| Eigenvalue | 2.526 | | | |
| % of variance explained | 31.575 | | | |

### 3.3.2. Sensitivity Index

Three variables greatly contributed to explaining the sensitivity of climate hazards: resource-dependency, accessibility to clean water during drought, and healthcare services. For resource-dependency, it was because households depended strongly on the fishery as a source of their revenue. With storms and typhoons, they were unable to operate their businesses, which caused a rapid decline in their income. Additionally, without sufficient water supply during the drought period, some families bought water at a relatively higher price while poor families managed to use water from wells. However, private wells cannot be dug deeper due to salt water, and the water was not clean enough to use, and often linked to health issues. The sensitivity degree was found to eb higher in Daun Tong and Steong Veng compared to Smach Meanchey, with indices of 0.856, 0.633, and 0.559, respectively (Table 3).

**Table 3.** Summary of sensitivity index.

| Descriptive | Component Score Coefficient | Daun Tong | Steong Veng | Smach Meanchey |
|---|---|---|---|---|
| | | Index | | |
| Damage to property and livestock due to climate hazards | 0.243 | 0.131 | 0.126 | 0.112 |
| Number of family member(s) injured due to floods, storms, and landslides | 0.266 | 0.001 | 0.000 | 0.000 |
| The level of household dependency on natural resources | 0.399 | 0.244 | 0.209 | 0.143 |
| Agricultural dependency for income | −0.202 | −0.012 | −0.047 | −0.043 |
| Road conditions after flooding | 0.071 | 0.034 | 0.028 | 0.030 |
| Distance from market (minutes of traveling) | 0.083 | 0.027 | 0.029 | 0.028 |
| Lack of clean water during drought | 0.383 | 0.258 | 0.141 | 0.146 |
| Accessibility to healthcare (level of receiving health service per year) | 0.375 | 0.159 | 0.147 | 0.142 |

**Table 3.** *Cont.*

| Descriptive | Component Score Coefficient | Daun Tong | Steong Veng | Smach Meanchey |
|---|---|---|---|---|
| | | Index | | |
| Total | | 0.856 | 0.633 | 0.559 |
| Eigenvalue | 1.708 | | | |
| % of variance explained | 21.348 | | | |

### 3.3.3. Adaptive Capacity Index

Table 4 indicates the relative importance of the role of income diversification, government assistance, and human capital, as high component scores were contributed by these factors. Overall, Daun Tong had the highest adaptive capacity with an index of 0.625 compared to the other two communes, with indices of 0.506 and 0.493 in Steong Veng and Smach Meanchey, respectively. The study found out that the adaptation in these three communes positively correlated to the rate of exposure with $r = 0.16$, $p < 0.1$. On the basis of interviews with many households in these three communes, it was shown that as households faced the low rate of exposure, they tended to have no intention to improve their adaption—for example, to buy extra equipment for self-protection or to renovate their house. However, correlation did not mean causation, and we cannot ignore the role of other factors that could also influence their correlation as well.

**Table 4.** Summary of adaptive capacity index.

| Descriptive | Component Score Coefficient | Daun Tong | Steong Veng | Smach Meanchey |
|---|---|---|---|---|
| | | Index | | |
| Housing quality | 0.095 | 0.050 | 0.046 | 0.049 |
| Tools and technology to access climate information (TV, radio, mobile phone) | 0.120 | 0.082 | 0.067 | 0.055 |
| Self-protection tools such as sandbags, life-jackets, and so on | 0.116 | 0.034 | 0.035 | 0.038 |
| Number of family members who finished grade 9 | 0.225 | 0.068 | 0.031 | 0.049 |
| Number of family members who earn income | 0.270 | 0.077 | 0.061 | 0.051 |
| Training or vocational course related to climate change attended by family members | 0.097 | 0.005 | 0.002 | 0.000 |
| Assistance from government | 0.295 | 0.063 | 0.028 | 0.007 |
| Availability of supportive policy | 0.220 | 0.037 | 0.004 | 0.008 |
| Diversification of income sources | 0.335 | 0.127 | 0.131 | 0.131 |
| Rice reserve during a shock | 0.169 | 0.012 | 0.016 | 0.014 |
| Ownership (animal, livestock) | 0.116 | 0.004 | 0.026 | 0.033 |
| Amount of borrowing from formal and informal sectors (debt: monthly) | 0.172 | 0.036 | 0.027 | 0.020 |
| Information sharing related to climate hazards with neighbors | 0.074 | 0.031 | 0.027 | 0.030 |
| Amount of social support from relatives and community during and after disaster | −0.066 | −0.011 | −0.008 | −0.008 |
| Dependency Ratio | 0.065 | 0.011 | 0.014 | 0.017 |
| Total | | 0.625 | 0.506 | 0.493 |
| Eigenvalue | 1.983 | | | |
| % of variance explained | 13.223 | | | |

3.3.4. Vulnerability Index

Table 5 demonstrates the climate vulnerability assessment of the three communes. Daun Tong was the most vulnerable region to climate hazards compared to Steong Veng and Smach Meanchey. This was due to its geographical location close to the sea, and the quality of housing was less durable and could be highly endangered by natural hazards. Moreover, the rise of seawater could cause health-related issues for households in Daun Tong due to infectious diseases, since households were located in a place that looked like a slum area. For Steong Veng, it was also similar to Daun Tong. One main reason that pushed them into this vulnerable condition was the households' construction. Instead of collectively concentrated, housing was isolated and far away from one from another, which would cause a higher risk of incurring damage from storms and typhoons.

**Table 5.** Contributing factors of climate vulnerability.

| Factor | Daun Tong | Steong Veng | Smach Meanchey |
|---|---|---|---|
| Exposure Index | 0.450 | 0.456 | 0.289 |
| Sensitivity Index | 0.856 | 0.633 | 0.559 |
| Adaptive Capacity Index | 0.625 | 0.506 | 0.493 |
| Vulnerability Index | 0.525 | 0.522 | 0.434 |

Figure 3 indicates the vulnerability triangle of the three communes. While the exposure rate was similar between Daun Tong and Steong Veng communes, in terms of adaptive capacity and sensitivity, Daun Tong held a relatively higher degree.

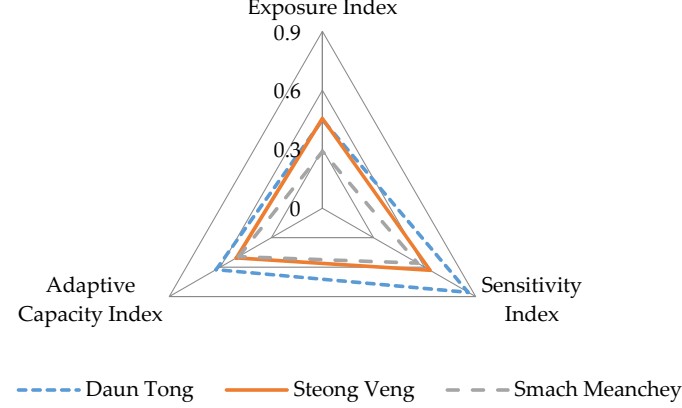

**Figure 3.** Vulnerability triangle diagram of contributing factors for Daun Tong, Steong Veng, and Smach Meanchey communes.

*3.4. Poverty and Vulnerability*

Table 6 reveals the vulnerability to poverty of poor and non-poor households by using the national and international poverty lines. In terms of the national poverty line, the result showed that, in total, less than 10% of households were poor while about three out of 10 households were facing vulnerability to poverty. One-third of poor households were vulnerable; on the other hand, only around one-fourth of non-poor were vulnerable to poverty. For the international poverty line, almost 20% of households were considered to be living in poverty with more than one-third of them facing vulnerability to poverty. Although both groups shared a similar proportion in terms of vulnerability,

slightly above one-third, the poor households tended to possess a bit higher proportion of people who were vulnerable to poverty compared to the non-poor households.[6]

**Table 6.** Vulnerability to poverty by using national and international poverty lines.

| Vulnerability Categories | National Poverty Line | | | International Poverty Line | | |
|---|---|---|---|---|---|---|
| | Proportion of Household (%) | | | Proportion of Household (%) | | |
| | Poor | Non-Poor | Poor and Non-Poor | Poor | Non-Poor | Poor and Non-Poor |
| Vulnerable V > 0.5 | 2 | 26 | 28 | 7 | 27 | 34 |
| Non-Vulnerable V ≤ 0.5 | 4 | 68 | 72 | 13 | 53 | 66 |
| Total | 6 | 94 | 100 | 20 | 80 | 100 |

Figures 4 and 5 identify the clusters of vulnerability and household income in the natural logarithmic form. The first one indicates the use of the national poverty line; the second applies the international poverty threshold. The area below the horizontal line is considered to be poor, while the area above is viewed as non-poor. On the other hand, the area on the left-hand side of the vertical line with vulnerability ≤ 0.5 is not considered to be vulnerable, while the right-hand side area represents the vulnerable households.

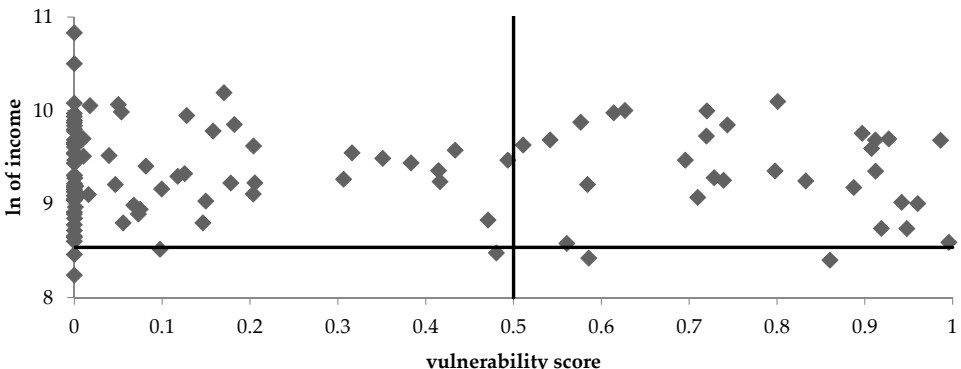

**Figure 4.** Natural log of income and vulnerability score using the national poverty line.

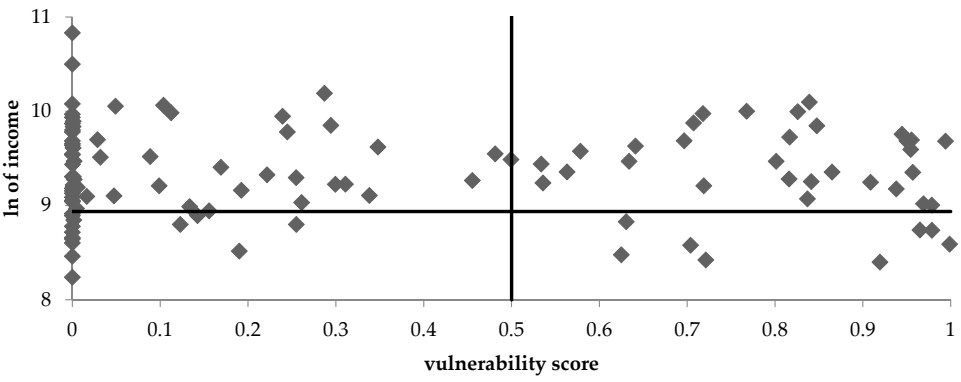

**Figure 5.** Natural log of income and vulnerability score using the international poverty line.

---

6   The result was only based on a one-time observation. The lack of experimental design may not capture the true vulnerability degree across time intervals and could also influence the result. The longitudinal study could be a better alternative method for a vulnerability assessment and be more flexible for the experiment. In addition, the use of low poverty thresholds and 50% cutoff point could also influence the result as well. As the study implemented low poverty thresholds, it may not represent the vulnerability in an urban area since the living cost was higher than the urban national poverty line. The use of the 50% cutoff point to indicate the state of being vulnerable in the next period could also lead to bias; however, the benchmark level was still subjective from the author's point of view.

## 4. Discussion

*Determinants of Vulnerability to Poverty*

The OLS regression result revealed the expected sign for most of the significant variables, except debt accessibility and household sizes, whereas the insignificant variables had a low value of β that was almost zero (Table 7).

**Table 7.** Determinants of vulnerability to poverty (OLS).

| Variables | National Poverty Line | | International Poverty Line | |
|---|---|---|---|---|
| | B | *t*-Value | B | *t*-Value |
| Age of respondent | −0.01 | −0.61 | 0.00 | 0.03 |
| Household size | −0.02 ** | −2.05 | −0.02 ** | −1.98 |
| Education of the head of household | −0.01 *** | −3.05 | −0.01 *** | −3.02 |
| Climate hazards (exposure index) | −0.01 | −0.26 | −0.01 | −0.12 |
| Agricultural dependency | 0.13 *** | 7.42 | 0.14 *** | 7.94 |
| Level of healthcare accessibility | −0.02 | −0.83 | −0.01 | −0.53 |
| Housing quality | −0.20 *** | −9.88 | −0.22 *** | −10.53 |
| Assistance from government during a shock | −0.05 | −1.05 | −0.04 | −0.84 |
| Income diversification | −0.18 *** | −5.41 | −0.22 *** | −6.47 |
| Possession of livestock asset | 0.10 ** | 2.48 | 0.12 *** | 3.01 |
| Debt accessibility | 0.22 *** | 6.74 | 0.24 *** | 7.44 |
| Access to information related to climate hazards | −0.06 *** | −4.65 | −0.06 *** | −4.51 |
| Constant | 1.07 *** | 10.47 | 1.10 *** | 10.61 |
| *F* | 33.26 | | 38.22 | |
| *R*-squared | 0.80 | | 0.82 | |
| Number of observations | 112 | | 112 | |

Note: *** $p < 0.01$, ** $p < 0.05$

Piya et al. (2012) explained that access to loans could be a social safety net to overcome all types of shocks. However, in this study, credit accessibility revealed an unexpected sign. In the case of borrowing money, households must pay the interest rate, and borrowing from the informal sector made it difficult to pay back due to high interest rates. A loan was good only if a household took it to invest or operate another business to generate extra income; in contrast, in Koh Kong, people borrow money for different purposes include housing renovation, daily consumption, and debt repayment. These activities could not improve the vulnerability to poverty, and will only make it worse in the next period. Using the national poverty line, the study found that a household that did have access to a loan had a vulnerability 0.22 higher compared to those that did not have access to a loan.

For household sizes, some studies indicated that an increase in family members would induce vulnerability, while others suggested that the nature of household members should be carefully considered because their influence depended on whether they were a dependent or a source of labor for the household. Ligon and Schechter (2003) found out that a large family would likely increase the dependency ratio, which would reduce the average income per household. However, if a greater number of the family members provided labor, then it would help the household to lower their vulnerability. Opiyo et al. (2014) revealed that household size contributed to reduce the vulnerability. The result of this study showed the negative impact of household size on the vulnerability degree by 0.02.

This study showed the expected results of the significant role of education, agricultural dependency, housing quality, income diversification, possessing of livestock, and information sharing.

For education, the study revealed that the education of the head of household could reduce the vulnerability to poverty. A one-year increase in formal education reduced the vulnerability by 0.01. The significant role of education has been recognized as a means for vulnerability reduction (Mendoza et al. 2014; Megersa 2015; Ncube et al. 2016). Moreover, this study found that the head of the family received formal education for only around five years on average, which is far below the national policy target, and that, typically, only one member of a family had finished 9th grade (Table 1).

Most of the household heads did not receive proper or qualified education during their adulthood. In this study, about two-thirds of respondents were female whose cultural constraint remained an obstacle; this situation still exists in many developing countries, including Cambodia.

Agricultural dependency could improve the situation of the household as it served as food security; however, in an urban area, a household that strongly depended on agriculture tended to be exposed to more risks, and it also limited their opportunity to work in other sectors. The result confirmed that agricultural dependency positively influenced the level of vulnerability. As stated in a cross-country study of Mendoza et al. (2014), that agriculture dependency is more exposed to climate-related disasters, especially drought for Cambodia. Ludeña and Yoon (2015) pointed out that agricultural dependency often led to social and economic stress.

Piya et al. (2012) identified that high livestock share was likely to increase sensitivity to climate hazards. In terms of the livestock assets, the study showed that possessing livestock resulted in a vulnerability level 0.10 higher than those who did not. Livestock was exposed to heat stress and disease. The sustained increase in temperature caused the scarcity of grassland. Furthermore, small-scale livestock assets made households pay less attention to their animal, so during prolonged flooding their livestock was sensitive to infectious diseases and ecological change.

Moser (1998) found that agricultural assets played a less important role for urban residents compared to human and physical assets such as housing. Chen et al. (2013) also revealed the significant role of housing quality to reduce vulnerability to natural hazards. The result from this study also showed that better housing quality contributed to lower vulnerability.

Although livelihood diversification seemed to be an effective method to strengthen adaptive capacity and to reduce vulnerability, it could also disrupt the skill and productivity of a household. Liu et al. (2008) emphasized that specialization played an important role in adaptation. Nonetheless, Andersen and Cardona (2013) indicated the crucial role of livelihood diversification as a strategy of resilience in their study on the comparison between urban and rural households in Bolivia. Kelly and Adger (2000) also emphasized that income diversification could lessen inequality and poverty. This study revealed that a diversified household had a lower rate of vulnerability compared to a non-diversified household.

High exposure and low adaptive capacity could be associated with a lack of knowledge and information of climate change. The sharing of information related to climate hazards played an important role, since many households depended on fishery activities. This study confirmed that the level of information sharing would contribute to the decline of vulnerability.

Gaiha and Imai (2008) found a negative impact of the age of the head of household on the level of vulnerability. The head of household tended to build up more experience to counter vulnerability as they grew older. Besides, Andersen and Cardona (2013) also emphasized that the more mature household had more time to accumulate wealth as an alternative source of their livelihood. This study showed the expected sign to support this argument, although it was not significant.

Mirza (2003) emphasized the crucial role of government intervention and coordination among relevant stakeholders to strengthen adaptive capacity in developing countries. In terms of government assistance, the result produced the expected sign, but it was not significant. To some extent, assistance reached the vulnerable group, although the size of government assistance was relatively small and could not help families recover from a shock.

The study indicated that healthcare accessibility could contribute to the decline of vulnerability to poverty. As Oni and Yusuf (2008) found that disease was the main agent in transforming non-vulnerable groups to vulnerable groups. Access to healthcare services could provide a human asset to improve a household's economic condition.

Lastly, the exposure index showed the unexpected sign and turned out to be not significant. However, in a correlation study, a positive relationship between climate-related exposure and vulnerability was found, so the result of the negative sign in the regression study could be due to the interaction among predictor variables in the study.

## 5. Conclusions

This study contributes to identify the most vulnerable location of climate change. To measure the climate vulnerability, we chose an index approach to compare climate vulnerability among three communes in Koh Kong. It was suggested that Daun Tong communes should be prioritized for intervention policies, as it experienced the highest climate vulnerability with an index of about 0.53 compared to the other two regions: Steong Veng and Smach Meanchey with indices of 0.52 and 0.43, respectively.

In addition, the study also investigates vulnerability in the context of poverty by perceiving vulnerability as the probability that a household income will fall below the poverty line. It revealed that, by using the international poverty line threshold, more than one-third of households have been facing vulnerability to poverty, while this value was only about 28% when the national poverty line was used. By using two poverty thresholds, the study showed that households were not sensitive to falling into the poverty. However, both poverty thresholds indicated that a higher proportion of poor families faced vulnerability to poverty compared to non-poor households.

By assuming that vulnerability arises as a result of shocks such as climate hazards and changes in household characteristics, the study found that improving education, income diversification, and physical assets such as housing could be effective methods to lower vulnerability. Lastly, borrowing should be done cautiously. This study pointed out that access to a loan may not be a wise decision as a household may be unable to pay back the loan with interest, and it induced a high vulnerability for the subsequent period.

## 6. Further Study

This study did not categorize the types of loans, whether a household borrowed from formal or informal sectors. Some studies suggested that access to authorized loans could improve the livelihood of a household, while borrowing from the informal sector would likely induce vulnerability. The next study should categorize the types of loan for a better comparison. Moreover, not all the variables that were highlighted were incorporated into the study due to methodological constraint. To illustrate, since households used various types of energy, the study could not capture it well, and it would lead to a bias to include it. Consultation with an expert is important to find good proxies that fit within the context.

Next, the use of PCA for weighing is effective; however, one issue is the correlation sign produced by PCA. For example, in the sensitivity study, the correlation signs of the road conditions and accessibility to healthcare were the opposite of what was expected. The next study is advised to conduct a rigorous study and consultation about the relationships among variables; otherwise, it would lead to a misinterpretation of the results.

Lastly, Since VEP works on many assumptions, some may not be fulfilled. One of these is heteroscedasticity of income. Whether income, consumption, or another variable should be used as a proxy is a crucial problem. In this study, income was used instead of consumption because it tended to vary across households more than consumption, and reflected the social status of the rich and poor households, so it was suitable for the assumption of heteroscedasticity. However, the record of household income and expenditure cannot avoid some errors due to the accessibility of data. The next study should implement a clear and standardized procedure to record household income and expenditure that is recommended by a formal institution in order to improve the validity of the data.

**Acknowledgments:** Author sincerely acknowledges the reviewers' comments and the supports from Chhinh Nyda and Thath Rido for their technical comments. Author also would like to show gratitude and appreciation to the Urban Climate Change Resilience in Southeast Asia (UCRSEA) project, which is funded by the Social Sciences and Humanities Research Council of Canada and the International Development Research Centre (IDRC) for funding support to conduct this research.

**Conflicts of Interest:** The author declares no conflict of interest.

**Appendix A**

Lists of indicators in this study were summarized in the 4 tables below. In totally, 8 variables were used to study the exposure degree (Table A1). It could be categorized into four main groups: flood, drought, Typhoon, and storm. Similarly, another 8 variables were constructed to measure the sensitivity (Table A2). 15 variables were used to assess the adaptive capacity in Table A3. In terms of vulnerability as expected poverty, a list of variables was summarized in Table A4.

**Table A1.** Indicators in exposure study; which the positive sign (+) refers to positive contribution to higher level of exposure.

| Exposure Indicators | Descriptive | Type of Measurement | Expected Sign |
|---|---|---|---|
| Flood | The number of floods affecting households that occurred during the last 5 years | Scale | + |
| | Impact of flooding on livelihood | Ordinal | + |
| Drought | Period of insufficient clean water usage per year during the last 5 years | Scale | + |
| | Impact of drought on livelihood | Ordinal | + |
| Typhoon | Frequency of typhoons affecting household livelihood per year during the last 5 years | Scale | + |
| | Impact of typhoons on livelihood | Ordinal | + |
| Storm | Frequency of storms affecting household livelihood per year during the last 5 years | Scale | + |
| | Impact of storms on livelihood | Ordinal | + |

**Table A2.** Indicators in sensitivity study; which the positive sign (+) refers to positive contribution to higher rate of sensitivity while the negative sign (−) refers to negative contribution to lower sensitivity degree.

| Descriptive | Type of Measurement | Expected Sign |
|---|---|---|
| Damage to property and livestock due to climate hazards | Ordinal | + |
| Number of family member(s) injured due to floods, storms, and landslides | Scale | + |
| The level of household dependency on natural resources | Ordinal | + |
| Agricultural dependency for income | Ordinal | + |
| Road conditions after flooding | ordinal | − |
| Distance from market (minutes of traveling) | Scale | + |
| Lacking clean water during drought | Ordinal | + |
| Accessibility to healthcare (level of receiving health services per year) | Ordinal | + |

**Table A3.** Indicators in adaptive capacity study; which the positive sign (+) refers to positive contribution to higher level of adaptive capacity while the negative sign (−) refers to negative contribution to lower adaptive capacity.

| Descriptive | Unit of Measurement | Expected Sign |
|---|---|---|
| Housing quality | Ordinal | + |
| Tools and technology to access climate information (TV, radio, mobile phone) | Ordinal | + |
| Self-protection tools such as sandbags, life-jackets, and so on | Scale | + |
| Number of family members who finished grade 9 | Scale | + |
| Number of family members who earn income | Scale | + |
| Training or vocational course related to climate change attended by family members | Scale | + |
| Assistance from government | Ordinal | + |

**Table A3.** *Cont.*

| Descriptive | Unit of Measurement | Expected Sign |
| --- | --- | --- |
| Availability of supportive policy | Ordinal | + |
| Diversification of income sources | Scale | + |
| Rice reserve during a shock | Ordinal | + |
| Ownership (animal, livestock) | Ordinal | + |
| Amount of borrowing from formal and informal sectors (debt: monthly) | Scale | + |
| Information sharing related to climate hazards with neighboring | Ordinal | + |
| Amount of social support from relatives and community during and after disaster | Ordinal | + |
| Dependency ratio | Ordinal | + |

**Table A4.** Variables in VEP study in which positive sign (+) refers to positive contribution to higher vulnerability while the negative sign (−) refers to negative contribution to lower vulnerability. For binary dummy variable, 0 indicates the absence while 1 indicates the presence of the effect, simply put it refers to (No, Yes) answer.

| Descriptive of Independent Variables | Unit of Measurement | Expected Sign |
| --- | --- | --- |
| Age of respondent | Scale (years) | − |
| Household sizes | Scale (persons) | + |
| Education of the head of household | Scale (years) | − |
| Climate hazards (exposure index) | Scale (Index) | + |
| Agricultural dependency | Ordinal (rating) | + |
| Level of healthcare accessibility | Ordinal (rating) | − |
| Housing quality | Ordinal (rating) | − |
| Assistance from government during a shock | Dummy (0.1) | − |
| Income diversification | Dummy (0.1) | − |
| Possession of livestock asset | Dummy (0.1) | + |
| Debt accessibility | Dummy (0.1) | − |
| Access to information related to climate hazards | Ordinal (rating) | − |

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
