# Peer review of "Urban Climate Vulnerability in Cambodia: A Case Study in Koh Kong Province"

_economies, doi:10.3390/economies5040041_

Round 1

Author Response

Summary
The paper uses an index-based approach to assess urban climate vulnerability in three different case studies in Cambodia and uses regression analysis to evaluate the association between vulnerability and poverty using both national and international poverty lines. Although, the paper is structured and written well, there are a several major issues which have to be addressed before the paper is mature for publication (see detailed comments below).
Major Compulsory Revisions
1. The paper is full of generic statements (e.g. page 1, lines 21-46); I think the authors should be more concrete and specific to the local idiosyncrasies in urban environments of Cambodia
Response: The generic statements were removed from the paper. I have narrowed it down to focus only in Cambodia context. I also changes some sentence structures as well.
2. P2, line 51: disasters are not natural, they result from the interaction of (socio-)natural hazards with the vulnerability of exposed elements (see IPCC 2014; O’Keefe et al. 1976, etc.)
Response: I am not quite clear about this comment. I would be appreciated if further explanation is provided for this point.
3. P2, line 81: don’t you want to study the vulnerability of urban residents towards climate change, rather than the vulnerability of CC on urban residents? Might be a language issue
Response: I find there is no difference between the two ideas. I want to study the vulnerability of climate change by comparing the three different regions by using index-based approach.
4. Index construction (P3, line 100 ff): it remains totally unclear
o (1) which definition of vulnerability (and associated conceptual framing) was applied and why;
Response: In fact, I followed the definition of IPCC in defining vulnerability as a function of exposure, sensitivity, and adaptive capacity. The reason that I followed this definition is because it is widely accepted and many literatures also considered it as a pioneer in vulnerability study.
o (2) which indicators were selected, how they were selected (e.g. systematic review of literature, expert consultations, etc.) and which datasets or proxies (incl. source, year, resolution, etc.) were used to represent these indicators; while I have no issues with the choices made, they need to be backed up scientifically and/or through expert knowledge; currently the choice seems to be arbitrary
Response: The indicators were selected through the review of the past literature, except the governance aspect. I found that some studies identified the important role of government to reduce the vulnerability, but the proxies were not discussed extensively, so I consulted with my supervisor in selecting the proxy for this indicator and came up with the Likert scale measurement.
o (3) how the household characteristics and the GINI coefficient described in section
3.1 and 3.2 respectively, are incorporated into the vulnerability assessment. One would expect that these are incorporated into the adaptive capacity index, but this is not the case
Response: household characteristic only to provide some information about the sample. It does not really incorporate into any part. Although it seems to stand alone, I feel that it is necessary here. For GINI index, the goal here is serving as the assumption of the income gap across the household. I think to study about the sensitivity of vulnerability to poverty, the fundamental of
this study is to assume that families in Koh Kong do not have a wider income gap,so it makes more sense when changing the poverty threshold.
o (4) were the data checked for outliers and multicollinearities? This should be included in the methods and results sections;
Response: the data were checked for outliers. For example, income was transformed by using the natural log to reduce the outlier issue. For colinearity, it also was checked as well. In case the collinearity among independent variables is too high, more than 0.7, one variable will be dropped out of the equation; however, none of the variables had correlation coefficient more than 0.6. Besides, it also was checked to see whether income or expenditure was more appropriate in the study. After the test, “income” variable was used instead of the “expenditure”.
5. Indicators (Tables 2, 3 and 4): some of the indicators need to be more specific; e.g. what is their measuring unit, do they increase or decrease vulnerability, etc (e.g. what does “magnitude of flooding on livelihood” or “availability of supportive policy” mean?, etc.). With the information that is currently provided it would be difficult to replicate the study; authors must be more specific, adhering to the quality criteria for indicators.
Response: I think it is about the language use. What I referred to is the level the flooding impact on their livelihood and also, supportive policies that household received as well. I used the Likert scale for these variables.
6. The vulnerability-poverty angle taken here is interesting, however the paper lacks a discussion on poverty as a root cause of vulnerability vs an outcome of “vulnerability” (according to the IPCC 2007 definition of vulnerability); I think the authors should discuss this more in depth; further it is not clear how the 2 definitions of vulnerability used here (i.e. IPCC 2007 vs Vulnerability as Expected Poverty) fit together – conceptually, empirically and methodologically
Response: The IPCC used in this study covered not only in the economic aspect. For the study of vulnerability as expected poverty was because I wanted to look how climate change affected the household condition economically. Will climate change likely to cause the household to be poor in the future? Somehow this paper did lack the discussion about the root cause of poverty Vs an outcome of vulnerability and also their relationship as well, but I could not address every aspect in one paper. However, I do really appreciate your comments to improve my writing.
7. Regression analysis: the authors should state more clearly which factors were used as dependent and independent variables and whether the assumptions for running an OLS regression were met.
Response: I think this point if looking at the regression table, one could tell which variable was dependent and which were independent variables. Running OLS was not to confirm the assumption, but rather to estimate the probability of being poor in the future. Although the OLS result was used in the discussion part, it did not serve as the main priority of this study.
8. Table 6, Fig 3 and 4: how was the threshold for vulnerability (0.5) defined?
Response: There is no clear cut of defining the vulnerability (0.5) since it is just a probability level. Some studies also used the probability up to 0.7 to be considered as vulnerability while some may define a threshold to categorize between low and high vulnerable. As the outcome only used two options as being vulnerable or non-vulnerable, so using 50-50 chance is appropriate.
Minor Compulsory Revisions
9. P1, lines 21-29: irrelevant information for this paper (Hong Kong, global fresh water resources, etc.). All these aspects are certainly relevant, but not associated with urban vulnerability in Cambodia
10. P1, lines 28-29: “one of its problems is its contribution to CC” I would refrain using such generic statements; authors should be more specific: how, why relevant here, etc.?
11. P1, lines 36-39: I agree that CC often (depends on the context) affects the urban poor more than better off populations in cities, however, the arguments provided here (alteration of ecosystems) are not the biggest issue for urban populations; the authors should be more careful with their argumentation
12. P2, lines 52-53: “rose to 10%” .. of what, all economic losses? Authors should be more precise.
13. P2, lines 59-60: I think you mean “are exposed to climate hazards” instead of “are at risk”.
14. Language: authors should ask a native speaker to proof-read the entire paper; it is currently not acceptable for publication
15. P2, lines: what does it mean that rainfall is predicted to rise from 2% to 6%? What is the baseline?
16. P2, line 88 (and throughout the manuscript): replace “consensus” with “census”
Response: All of these errors were resolved. I edited on misspelling, grammar, and sentence structure. Besides, it was also changed for a better consistency.

Reviewer 2 Report

This paper adds to the body of empirical studies on climate vulnerability, focusing on an urban area in coastal Cambodia. This is an area that is both vulnerable to climate change impacts and understudied. The main methodological contribution is through the construction and application of the vulnerability index, linking vulnerability to poverty.

Given the focus of the paper, I would expect it to be better anchored in literature on vulnerability and social, economic and political factors (see, e.g., Pelling, 2003, 2011; Wisner et al., 2004). It would be important for the authors to place the article better into the context of existing vulnerability literature.

I’d also like to bring to the authors’ attention a related study in Cambodia, which would be of direct interest from a methodological point of view (Halsnaes, Kaspersen & Traerup, 2016).

Given the place-based nature of the article, I would very much like to see a map of the location of the study area and, probably, more detailed maps related to the findings.

Another aspect that will need attention is language. There are lots of problems with syntax and some sentences are incomplete or not readily understandable.

References:

Halsnaes, K., Kaspersen, P.S. & Traerup, S. (2016). Climate change risk – Methodological framework and case study of damages from extreme events in Cambodia. In: Uitto, J.I. & Shaw, R. (eds.), Sustainable development and disaster risk reduction, pp. 71-85. Tokyo: Springer.

Pelling, M. (2003). The vulnerability of cities: Natural disasters and social resilience. Oxon: Earthscan.

Pelling, M. (2011). Adaptation to climate change: From resilience to transformation: New York: Routledge.

Wisner, B, Blaikie, P., Cannon, T. & Davis, I. (2004). At risk: Natural hazards, people’s vulnerability and disasters. London: Routledge.

Author Response

This paper adds to the body of empirical studies on climate vulnerability, focusing on an urban area in coastal Cambodia. This is an area that is both vulnerable to climate change impacts and understudied. The main methodological contribution is through the construction and application of the vulnerability index, linking vulnerability to poverty.
Given the focus of the paper, I would expect it to be better anchored in literature on vulnerability and social, economic and political factors (see, e.g., Pelling, 2003, 2011; Wisner et al., 2004). It would be important for the authors to place the article better into the context of existing vulnerability literature.
Response: I would take this comment into consideration. I have included some points for the existing vulnerability literature on building the vulnerability index in the revised version.

I’d also like to bring to the authors’ attention a related study in Cambodia, which would be of direct interest from a methodological point of view (Halsnaes, Kaspersen & Traerup, 2016).
Response: I think I already went through some studies about the vulnerability study in Cambodia including the methodology that was used in different parts of Cambodia. Since there are constraints in accessibility and limited study in the urban area of Cambodia, the information may be still lacking to some extent.

Given the place-based nature of the article, I would very much like to see a map of the location of the study area and, probably, more detailed maps related to the findings.
Response: I included the map of the study area in the methodology part, but I am not so clear what you mean by detailed maps related to the findings. I hope to get a clarification if possible.

Another aspect that will need attention is language. There are lots of problems with syntax and some sentences are incomplete or not readily understandable.
Response: I edited and corrected this problem.

References:
Halsnaes, K., Kaspersen, P.S. & Traerup, S. (2016). Climate change risk – Methodological framework and case study of damages from extreme events in Cambodia. In: Uitto, J.I. & Shaw, R. (eds.), Sustainable development and disaster risk reduction, pp. 71-85. Tokyo: Springer.
Pelling, M. (2003). The vulnerability of cities: Natural disasters and social
resilience. Oxon: Earthscan.
Pelling, M. (2011). Adaptation to climate change: From resilience to transformation: NewYork:Routledge.
Wisner, B, Blaikie, P., Cannon, T. & Davis, I. (2004). At risk: Natural hazards, people’s vulnerability and disasters. London: Routledge.

Reviewer 3 Report

In this article, the authors assess the climate vulnerability of three urban communes in Cambodia. The authors create an index variable based upon “vulnerability as expected poverty”, with regards to exposure, sensitivity, and adaptive capacity of the household.

The paper needs significant revision for clarity in many ways. The introduction needs heavy revision to focus the intent of the paper, and for clarity of English. The first sentence, for example, talks about global fresh water resources, which is hardly related to the overall paper.

In the methodology, the paper needs start by explaining the reasoning behind creating an index, and then move in details of the components and process (such as the normalization section, which comes out of context and order).

Very important to include is the sampling method. How were these communes chosen? Then, how were the households chosen. Without context on the sampling, we have no way to know if these results are representative. Further, in this context, it is very difficult to compare the three communes as the core result. The description of these observations is missing from the data.

Following this, a better contribution would be to analyze what components of the index best explain vulnerability. What we learn in this situation by comparing the Cambodian and World Bank thresholds? Is one better than the other in representing true vulnerability?

Please provide better organization and interpretation of the regression.

With substantial revision, this paper may be a useful contribution to the literature.

Specific comments and example errors:

Line 4: What is “implying index approach”? Is this correct?

Line 12: adjusting for

Line 16: grammatical problems

Line 47: What does “Cambodia is not an exceptional case.” mean?

Line 87: What is “National consensus”?

Line 157: subject missing

Line 165: does dropping negative values generate a bias

Line 282: grammatical problems

Line 373: grammatical problems/unclear wording

Author Response

In this article, the authors assess the climate vulnerability of three urban communes in Cambodia. The authors create an index variable based upon “vulnerability as expected poverty”, with regards to exposure, sensitivity, and adaptive capacity of the household.
The paper needs significant revision for clarity in many ways. The introduction needs heavy revision to focus the intent of the paper, and for clarity of English. The first sentence, for example, talks about global fresh water resources, which is hardly related to the overall paper.
Response: Thank for the comment. I revised the introduction to focus only in Cambodia context and irrelevant information was withdrawn from the article.
In the methodology, the paper needs start by explaining the reasoning behind creating an index, and then move in details of the components and process (such as the normalization section, which comes out of context and order).
Response: The reasoning has been added to the new version. Creating index is to be able to compare across regions. It is to create the uniform dataset or a process to make standardized data.
Very important to include is the sampling method. How were these communes chosen? Then, how were the households chosen. Without context on the sampling, we have no way to know if these results are representative. Further, in this context, it is very difficult to compare the three communes as the core result. The description of these observations is missing from the data.
Response: The sampling method has been added to the new version, and also how the sample is selected. I have no idea if my study represents or be able to generalize. The way I choose this area is because it fits into the definition of the urban area and wants to study it as a case study in Cambodia.
Following this, a better contribution would be to analyze what components of the index best explain vulnerability. What we learn in this situation by comparing the Cambodian and World Bank thresholds? Is one better than the other in representing true vulnerability?
Response: The the component score of coefficient could tell which variables best explain each element; however, it is not the interest of this paper. Based on this study, it is not sufficient to tell which component best explain vulnerability. As mentioned in the paper, using two different thresholds is to study the sensitivities of household income level to poverty. Since not many households fall into the poverty, it showed that in the study area, households are not living so close to the poverty level. It is not to
explain that one threshold is better than the other.
Please provide better organization and interpretation of the regression.
Response: I have revised the interpretation of the result and edited on some parts.
With substantial revision, this paper may be a useful contribution to the literature. Specific comments and example errors:
Line 4: What is “implying index approach”? Is this correct? Line 12: adjusting for
Line 16: grammatical problems
Line 47: What does “Cambodia is not an exceptional case.” mean? Line 87: What is “National consensus”?
Line 157: subject missing
Line 165: does dropping negative values generate a bias Line 282: grammatical problems
Line 373: grammatical problems/unclear wording
Response: I have changed all the grammatical error and sentence structure, including wrong word and spelling.

Round 2

Reviewer 1 Report

The authors have addressed most of the comments and delivered a point-by-point response. While I agree with most of their responses and associate revisions, a few additional issues have to be addressed before the paper can be considered mature for publication: 

Major revisions:

Line 35 & line 274: Disasters are not natural! They are not only the result of natural hazards, but result from the interaction of hazardous events with vulnerable elements (see for example latest definition of the IPCC AR5 WGII or the IPCC SREX report). I have already flagged this in the initial review. I think the authors mean "hazard" when they mention "natural disaster". Care has to be taken with the terminology. 

Lines 41-42: the authors mention waste and sanitation, water quality and energy security as issues contributing to climate change vulnerability in Cambodia, however, apart from "lack of clean water during drought", none of these factors is included in the vulnerability assessment present here (see also appendix B). The authors should explain why they did not consider these indicators if they were highlighted by previous studies to be important determinants. 

Section 2.2 (lines 88ff): it is not clear (1) when the survey was conducted, where and how, (2) how many households were interviewed, and (3) how many samples were selected in each of the 3 communes. The authors should provide more details. Also an ethics statement should be provided since primary data was collected and analyzed.

Line 97ff: The authors use the old IPCC AR4 definition of vulnerability (IPCC 2007), but list the new IPCC AR5 WGII report (IPCC 2014) as a source. The way the authors conceptualize vulnerability and operationalize it in their assessment follows the logic and terminoligy of the IPCC 2007 report. Hence the authors should be clear about that and cite the appropriate report. 

Line 121: the authors mention the drawbacks of using PCA for obtaining indicator weights, but then conclude that "the problem would not be serious" and that "it was recommended to just ignore the issue". The authors should describe the problem more clearly, incl why it would not be serious in this specific case and why it was decided to "ignore the issue". More information is needed here. 

Line 189 ff: the authors run linear OLS regression, but no information is presented on whether or not the assumptions for a linear regression were met (e.g. linear relationship between outcome and predictor variables, normal distribution of residuals, etc.). If the assumptions were not met, the outcomes of the regression analysis might not be reliable. The authors should provide more details. I have raised this before, but I am not satisfied with the response from the authors (that it was not the "main priority of this study" and hence they did not care about assumptions). The authors refer to the outcomes of the OLS regression in the discussion (e.g. lines 309, etc.)

Line 397-399: the authors state that "poor families are vulnerable to poverty compared to non-poor households". This does not make sense. 

Minor revisions:

The entire paper needs to be carefully checked regarding spelling/grammar issues (especially the parts which have been added following the first review). There are still a few minor mistakes here and there (incl. in the abstract). 

Lines 22-28: while I agree with what has been added here, it is not clear whether this is a generic statement (i.e. applying to urban environments across the globe) or specific to the situation in Cambodia. The authors need to be more specific here and should add references. 

Lines 31-32: the authors should give an example how "urbanization has induced climate-related problems" in Cambodia.

Figure 1: needs a heading

Lines 106-115: the authors should only include those studies that are thematically related (urban vulnerability to climate change). Several important (recent) studies on urbanization, vulnerability and climate change are not included in the list presented here.

Line 118: I would disagree that weighing is the "MAIN" challenge in index construction. It is one of many challenges in index construction.

Line 223  & 224: currencies should also be given in USD. Otherwise readers have to convert. 

Table 1: is this an average across the study area? Are there differences across the three communes? Not clear. 

Lines 229-233: this should go into methods

Lines 349ff: why is it that high livestock share increases urban vulnerability to climate change?

Author Response

Major revisions:

Line 35 & line 274: Disasters are not natural! They are not only the result of natural hazards, but result from the interaction of hazardous events with vulnerable elements (see for example latest definition of the IPCC AR5 WGII or the IPCC SREX report). I have already flagged this in the initial review. I think the authors mean "hazard" when they mention "natural disaster". Care has to be taken with the terminology. 

I change it in the revised version. You are correct I referred it as the natural hazards.

Lines 41-42: the authors mention waste and sanitation, water quality and energy security as issues contributing to climate change vulnerability in Cambodia, however, apart from "lack of clean water during drought", none of these factors is included in the vulnerability assessment present here (see also appendix B). The authors should explain why they did not consider these indicators if they were highlighted by previous studies to be important determinants. 

So far, I have no idea how to observe these variables. In terms of waste and sanitation, I used healthcare instead as these two variables often link to health care issue. For energy security, since household used different types of energy and some of them could not be captured, it would cause a bias in the result, so I did not use it as well. I put it as the limitation of this study and suggested for future study.

Section 2.2 (lines 88ff): it is not clear (1) when the survey was conducted, where and how, (2) how many households were interviewed, and (3) how many samples were selected in each of the 3 communes. The authors should provide more details. Also an ethics statement should be provided since primary data was collected and analyzed.

I have edited and included the information in the sampling methodology.

Line 97ff: The authors use the old IPCC AR4 definition of vulnerability (IPCC 2007), but list the new IPCC AR5 WGII report (IPCC 2014) as a source. The way the authors conceptualize vulnerability and operationalize it in their assessment follows the logic and terminoligy of the IPCC 2007 report. Hence the authors should be clear about that and cite the appropriate report. 

As I format my computer, I have lost the source of references. I am appreciated for this comment as it is important to me. I changed the reference and used the original source as my reference.

Line 121: the authors mention the drawbacks of using PCA for obtaining indicator weights, but then conclude that "the problem would not be serious" and that "it was recommended to just ignore the issue". The authors should describe the problem more clearly, incl why it would not be serious in this specific case and why it was decided to "ignore the issue". More information is needed here. 

I included more information on this issue, the reason that the sign is not the issue (line 139-145).

Line 189 ff: the authors run linear OLS regression, but no information is presented on whether or not the assumptions for a linear regression were met (e.g. linear relationship between outcome and predictor variables, normal distribution of residuals, etc.). If the assumptions were not met, the outcomes of the regression analysis might not be reliable. The authors should provide more details. I have raised this before, but I am not satisfied with the response from the authors (that it was not the "main priority of this study" and hence they did not care about assumptions). The authors refer to the outcomes of the OLS regression in the discussion (e.g. lines 309, etc.)

For this methodology, VEP, it has its own assumption, because the OLS procedure could not produce efficient estimator that why it needed to transform the data by using weigh OLS (FGLS). I have followed these steps and confirm with the residuals test. I did not put this information because I think it will produce overload information for the reader. It is the same for the regression result in OLS that I put in discussion part,  I used the same residual test to make that the residual was not correlated with the explanatory variables (independent).

Line 397-399: the authors state that "poor families are vulnerable to poverty compared to non-poor households". This does not make sense. 

Because VEP runs on the probability, we are not sure if the poor or non-poor household had a higher probability rate. It does not mean that the poor always had a higher vulnerability. It could be the way I express my idea is not well organized, so I tried to rephrase it I the revised version.

Minor revisions:

The entire paper needs to be carefully checked regarding spelling/grammar issues (especially the parts which have been added following the first review). There are still a few minor mistakes here and there (incl. in the abstract). 

I corrected this issue one more time, please see it again in the revised version.

Lines 22-28: while I agree with what has been added here, it is not clear whether this is a generic statement (i.e. applying to urban environments across the globe) or specific to the situation in Cambodia. The authors need to be more specific here and should add references. 

This part is the general information, I included a few references in the revised manuscript.

Lines 31-32: the authors should give an example how "urbanization has induced climate-related problems" in Cambodia.

I have added a few examples in Cambodia case.

Figure 1: needs a heading

Lines 106-115: the authors should only include those studies that are thematically related (urban vulnerability to climate change). Several important (recent) studies on urbanization, vulnerability and climate change are not included in the list presented here.

I already addressed the issue and changed some parts as necessary.

Line 118: I would disagree that weighing is the "MAIN" challenge in index construction. It is one of many challenges in index construction.

Although I agree that there are many challenges on constructing the index, in my case, weighing is the main challenge. Maybe it is a different case for a different author, so I changed it as a request.

Line 223  & 224: currencies should also be given in USD. Otherwise readers have to convert. 

I changed it as requested; however, if possible, I preferred to keep it as local currency.

Table 1: is this an average across the study area? Are there differences across the three communes? Not clear. 

I added extra information on this one, although I did not give more explanation on this. In fact, I tested the information on income and expenditure across communes. The information was quite similar. Only income of households in Daun Tong commune was significantly different from the rest while the other variables were not significantly different.

Lines 229-233: this should go into methods

The information here was too short. If I put it into the methodology, it would stand alone. I preferred to keep it here.

Lines 349ff: why is it that high livestock share increases urban vulnerability to climate change?

I put an extra explanation in the revised version, please refer to it for more information.

Reviewer 3 Report

These edits improve the paper, but there remain many problems. Most of all, it is unclear how the methods address objective 1 "To identify the most vulnerable region of climate change," if the sample is not representative. The paper needs to be clarified as to what the comparison reveals. While the sampling text is more forthcoming, a convenience sample still needs to be justified, and explain what it means to select on "geographical condition." There are still a substantial number of language mistakes, included some new ones in the revisions, which reduce the clarity of the paper.

Author Response

Your concern may be on the sampling. I intended to compare the vulnerability not to choose the samples that represent the population.

By geographical condition, I refer to the fact that based on its past experience that location tends to face flooding and storm or more likely drought. I am not sure if it will be biased this way, but I think it will be reliable for comparison. In terms of convenience sampling, it may be facing a small issue in the study. As a small percentage of households could not be accessed and not be available, I chose the neighboring in the same community as the substitution.

I corrected the language mistakes in the revised version.

I could not understand or address all the problems you concern properly if you just mention “language mistakes” or “many problems”. Please point out specifically.